# A Weighted Mini-Bucket Bound Heuristic for Solving Influence Diagrams

## Abstract

Influence diagrams provide a modeling and inference framework for sequential decision problems, representing the probabilistic knowledge by a Bayesian network and the preferences of an agent by utility functions over the random variables and decision variables. MDPs and POMDPS, widely used for planning under uncertainty can also be represented by influence diagrams. The time and space complexity of computing the maximum expected utility (MEU) and its maximizing policy is exponential in the induced width of the underlying graphical model, which is often prohibitively large due to the growth of the information set under the sequence of decisions. In this paper, we develop a weighted mini-bucket approach for bounding the MEU. These bounds can be used as a stand-alone approximation that can be improved as a function of a controlling i-bound parameter. They can also be used as heuristic functions to guide search, especially for planning such as MDPs and POMDPs. We evaluate the scheme empirically against state-of-the-art, thus illustrating its potential.

## Introduction

An influence diagram (ID) (Howard and Matheson 2005) is a graphical model for sequential decision-making under uncertainty that compactly captures the local structure of the conditional independence of probability functions and the additivity of utility functions. Its structure is captured by a directed acyclic graph (DAG) over nodes representing the variables (decision and chance variables). The standard query on an ID is finding the maximum expected utility (MEU) and the corresponding optimal policy for each decision, subject to the history of observations and decisions.

Computing the MEU is recognized as one of the hardest tasks over graphical models, and hence recent work aims at developing anytime bounding schemes that tighten the bounds if more time and memory is available. Often, the target is to incorporate such bounds as heuristic functions to guide search algorithms. In this paper, we focus on computing the upper bound of MEU for a single agent sequential decision making problem with no-forgetting assumptions. We build on the methodology of weighted mini-bucket with cost-shifting that was used in the past for bounding probabilistic queries such as the partition function, Maximum A Posteriori (MAP) and Marginal MAP (MMAP) (Dechter and Rish 2003;

Liu and Ihler 2012; Ihler et al. 2012; Marinescu, Dechter, and Ihler 2014; Marinescu et al. 2018).

## Earlier Work

Upper bounding schemes for IDs are mostly based on either decomposing the underlying graphical model of an ID or on relaxing the information constraints that impose a partial elimination ordering for inference (e.g., by variable elimination). Both approaches are orthogonal and both can contribute to tightening the bounds. We elaborate below.

**Decomposition-based bounds** A preliminary extension of the mini-bucket scheme (Dechter and Rish 2003) to the MEU tasks was presented in a workshop paper by (Dechter 2000). This scheme decomposes the constrained join-tree (Jensen, Jensen, and Dittmer 1994) to a mini-bucket tree by partitioning a cluster in the join-tree into mini-buckets whose number of variables is bounded by the $i$-bound parameter. The mini-bucket scheme outputs bounds conditioned on partial assignment relative to a variable ordering and is therefore well poised to yield heuristics for search along the same ordering. Also recently, dual decomposition schemes, which are not directional, were extended for mixed graphical models queries such as Marginal Map (MMAP) by (Ping, Liu, and Ihler 2015). The scheme was also extended to the MEU task by (Lee, Ihler, and Dechter 2018) using the framework of *valuation algebra* (Shenoy and Shafer 1990; Mauá, de Campos, and Zaffalon 2012; Moral 2018). The valuation algebra for IDs defines operators such as combination and marginalization over pairs of probability and utility functions called *potentials* (Jensen, Jensen, and Dittmer 1994).

**Information relaxation** An alternative approach is to relax the information constraints in the ID thus allowing flexible variable orderings for processing (Nilsson and Hohle 2001). In particular, the information relaxation scheme for IDs can be viewed as re-ordering the chance variables in the constrained variable ordering. (Yuan, Wu, and Hansen 2010) integrated the re-ordering upper bounds as heuristics to guide a branch and bound search algorithm for solving IDs.

## Contributions

We develop a weighted mini-bucket scheme for generating upper bounds on the MEU. Given a consistent variable ordering, the scheme generates bounds for each variable conditioned on past histories, observations and decisions relative to a given variable ordering. We show empirically that the scheme can offer effective bounds faster than current state-of-the-art schemes. Thus, these bounds have high potential to be used as heuristics for search, in future work.

## Background

### Influence Diagrams

An ID is a tuple $\mathcal{M} := \langle \mathbf{C}, \mathbf{D}, \mathbf{P}, \mathbf{U}, \mathcal{O} \rangle$ consisting of a set of discrete random variables $\mathbf{C} = \{C_i | i \in \mathcal{I}_{\mathbf{C}}\}$, a set of discrete decision variables $\mathbf{D} = \{D_i | i \in \mathcal{I}_{\mathbf{D}}\}$, a set of conditional probability functions $\mathbf{P} = \{P_i | Pi \in \mathcal{I}_{\mathbf{P}}\}$, and a set of real-valued additive utility functions $\mathbf{U} = \{U_i | Ui \in \mathcal{I}_{\mathbf{U}}\}$. We use $\mathcal{I}_{\mathbf{S}} = \{0, 1, \cdots, |S| - 1\}$ to denote the set of indices of each element in a set $\mathbf{S}$, where $|S|$ is the cardinality of $\mathbf{S}$. As illustrated in Figure 1, an ID can be associated with a DAG containing three types of nodes: the chance nodes $\mathbf{C}$ drawn as circles, the decision nodes $\mathbf{D}$ drawn as squares, and the value nodes $\mathbf{U}$ drawn as diamonds. There are also three types of directed edges: edges directed into a chance node $C_i$ from its parents $pa(C_i) \subseteq \mathbf{C} \cup \mathbf{D}$ representing the conditional probability function $P_i(C_i | pa(C_i))$, edges directed into a value node $U_i$ denoting the utility function $U_i(\mathbf{X}_i)$ from its scope $\mathbf{X}_i \subseteq \mathbf{C} \cup \mathbf{D}$, and informational arcs (dashed arrows in Figure 1) directed from chance nodes to a decision node. The set of parent nodes associated with a decision node $D_i$ is called the information set $I_i$, and denotes chance nodes that are assumed to be observed immediately before making decision $D_i$. The constrained variable ordering $\mathcal{O}$ obeys a partial ordering which alternates between information sets and decision variables $\{\mathbf{I}_0 < D_0 < \cdots < \mathbf{I}_{|\mathbf{D}|-1} < D_{|\mathbf{D}|-1} < \mathbf{I}_{|\mathbf{D}|}\}$. The partial elimination ordering should ensure the regularity of the ID (a decision can only be preceded by at most one decision), and dictates the available information at each decision $D_i$ so that the *non-forgetting agent* makes decisions in a multi-staged manner based on the history available at each stage $i$, $H(D_i) := \cup_{k=0}^{i} \{D_k\} \cup \cup_{k=0}^{i} \mathbf{I}_i$. Solving an ID is computing the maximum expected utility $E[\sum_{U_i \in \mathbf{U}} U_i | \Delta]$ and finding a set of optimal policies $\Delta = \{\Delta_i | \Delta_i : R(D_i) \mapsto D_i, \forall D_i \in \mathbf{D}\}$, where $\Delta_i$ is a deterministic decision rule for $D_i$ and $R(D_i) \subseteq H(D_i)$ is the subset of history called the *requisite information* to $D_i$, namely, the only relevant history for making a decision (Nielsen and Jensen 1999).

### Valuation Algebra

The valuation algebra for IDs is an algebraic framework for computing the expected utility values, or values for short, based on the combination and marginalization on potentials (Jensen, Jensen, and Dittmer 1994; Mauá, de Campos, and Zaffalon 2012; Lee, Ihler, and Dechter 2018). Let a valuation $\Psi(\mathbf{X})$ be a pair of probability and value functions $(P(\mathbf{X}), V(\mathbf{X}))$ over a set of variables $\mathbf{X}$ called its scope. Occasionally, we will abuse the notation by dropping the

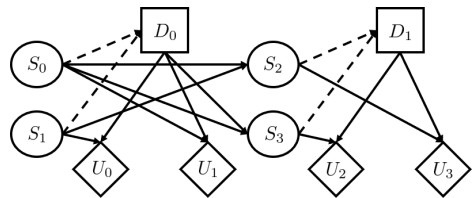

Figure 1: Factored MDP as Influence Diagram

scope from a function, e.g., writing $P_1(\mathbf{X}_1)$ as $P_1$. The combination and marginalization operators are defined as follows.

**Definition 1.** *(combination of two valuations)* *Given two valuations* $\Psi_1(\mathbf{X}_1) := (P_1(\mathbf{X}_1), V_1(\mathbf{X}_1))$ *and* $\Psi_2(\mathbf{X}_2) := (P_1(\mathbf{X}_2), V_1(\mathbf{X}_2))$, *the combination of the two valuations over* $\mathbf{X}_1 \cup \mathbf{X}_2$ *is defined by*

$$\Psi_1(\mathbf{X}_1) \otimes \Psi_2(\mathbf{X}_2) := (P_1 P_2, P_1 V_2 + P_2 V_1).$$

Following Definition 1, the identity is $(1, 0)$ and the inverse of $(P(\mathbf{X}), V(\mathbf{X}))$ is $(1/P(\mathbf{X}), -V(\mathbf{X})/P^2(\mathbf{X}))$.

**Definition 2.** *(marginalization of a valuation)* *Given a valuation* $\Psi(\mathbf{X}) := (P(\mathbf{X}), V(\mathbf{X}))$, *marginalizing over* $\mathbf{Y} \subseteq \mathbf{X}$ *by summation, maximization, or powered-summation with weights* $\mathbf{w}$ *are defined by*

$$\sum_{\mathbf{Y}} \Psi(\mathbf{X}) := (\sum_{\mathbf{Y}} P(\mathbf{X}), \sum_{\mathbf{Y}} V(\mathbf{X})),$$

$$\max_{\mathbf{Y}} \Psi(\mathbf{X}) := (\max_{\mathbf{Y}} P(\mathbf{X}), \max_{\mathbf{Y}} V(\mathbf{X})),$$

$$\sum_{\mathbf{Y}}^{\mathbf{w}} \Psi(\mathbf{X}) := (\sum_{\mathbf{Y}}^{\mathbf{w}} P(\mathbf{X}), \sum_{\mathbf{Y}}^{\mathbf{w}} V(\mathbf{X})).$$

The powered-sum elimination operator $\sum_X^w$ is defined by $\sum_X^w f(X) = [\sum_X |f(X)|^{1/w}]^w$, which replaces maximization and summation with a weight $0 \leqslant w \leqslant 1$ for a variable $X$, and it reduces to maximization and summation when $w \to 0$ and $w = 1$, respectively.

Finally, we define the comparison operator for the valuation algebra as a partial order as follows.

**Definition 3.** *(comparison of two valuations)* *Given two valuations* $\Psi_1 := (P_1, V_1)$ *and* $\Psi_2 : (P_2, V_2)$ *on the same scopes, we define inequality* $\Psi_1 \leqslant \Psi_2$ *iff.* $P_1 \leqslant P_2$ *and* $V_1 \leqslant V_2$.

In the following section, we state the decomposition bound using the valuation algebra notation, which were defined in (Lee, Ihler, and Dechter 2018).

### Decomposition Bounds

An ID can be compactly represented by the valuation algebra as $\mathcal{M} := \langle \mathbf{X}, \mathbf{\Psi}, \mathcal{O} \rangle$, where $\mathbf{X} = \mathbf{C} \cup \mathbf{D}$ and $\mathbf{\Psi} = \{(P_i, 0) | P_i \in \mathbf{P}\} \cup \{(1, U_i) | U_i \in \mathbf{U}\}$. The MEU can be written as

$$\sum_{\mathbf{I}_0} \max_{D_0} \cdots \sum_{\mathbf{I}_{|\mathbf{D}|-1}} \max_{D_{|\mathbf{D}|-1}} \sum_{\mathbf{I}_{|\mathbf{D}|}} \bigotimes_{\alpha \in \mathcal{I}_{\Psi}} \Psi_\alpha(\mathbf{X}_\alpha), \qquad (1)$$

where $\mathbf{X}_\alpha$ denotes the scope of $\Psi_\alpha$. The dependence relation between variables can be captured by a primal graph

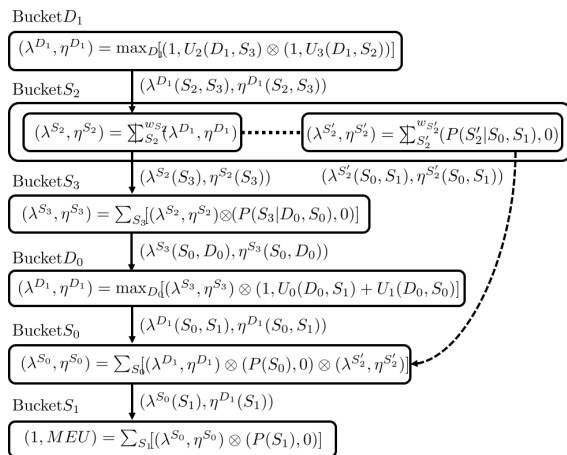

Bucket $D_1$

$(\lambda^{D_1}, \eta^{D_1}) = \max_{D_1}[(1, U_2(D_1, S_3) \otimes (1, U_3(D_1, S_2))]$

Bucket $S_2$     $(\lambda^{D_1}(S_2, S_3), \eta^{D_1}(S_2, S_3))$

$(\lambda^{S_2}, \eta^{S_2}) = \sum_{S_2}^{w_{S_2}} (\lambda^{D_1}, \eta^{D_1})$     $(\lambda^{S_2'}, \eta^{S_2'}) = \sum_{S_2'}^{w_{S_2'}} (P(S_2'|S_0, S_1), 0)$

Bucket $S_3$     $(\lambda^{S_2}(S_3), \eta^{S_2}(S_3))$     $(\lambda^{S_2'}(S_0, S_1), \eta^{S_2'}(S_0, S_1))$

$(\lambda^{S_3}, \eta^{S_3}) = \sum_{S_3}[(\lambda^{S_2}, \eta^{S_2}) \otimes (P(S_3|D_0, S_0), 0)]$

Bucket $D_0$     $(\lambda^{S_3}(S_0, D_0), \eta^{S_3}(S_0, D_0))$

$(\lambda^{D_1}, \eta^{D_1}) = \max_{D_0}[(\lambda^{S_3}, \eta^{S_3}) \otimes (1, U_0(D_0, S_1) + U_1(D_0, S_0))]$

Bucket $S_0$     $(\lambda^{D_1}(S_0, S_1), \eta^{D_1}(S_0, S_1))$

$(\lambda^{S_0}, \eta^{S_0}) = \sum_{S_0}[(\lambda^{D_1}, \eta^{D_1}) \otimes (P(S_0), 0) \otimes (\lambda^{S_2'}, \eta^{S_2'})]$

Bucket $S_1$     $(\lambda^{S_0}(S_1), \eta^{D_1}(S_1))$

$(1, MEU) = \sum_{S_1}[(\lambda^{S_0}, \eta^{S_0}) \otimes (P(S_1), 0)]$

Figure 2: Weighted Mini-Bucket Elimination example. We show a mini-bucket tree decomposition of the ID in Figure 1. The mini-bucket tree was generated along a constrained elimination ordering $D_1, S_2, S_3, D_0, S_0, S_1$ (from top to bottom) of the variables labelling the buckets. The mini-buckets are created at bucket $S_2$ by limiting the maximum cluster size from 4 to 3 ($i$-bound is 3). The non-negative weights $w_{S_2}$ and $w_{S_2'}$ for the variable $S_2$ associated with mini-buckets $S_2$ and $S_2'$ sum to 1.

$\mathcal{G}_p = (V, E)$, where the set of nodes $V$ are the variables, and an edge $e \in E$ connects two nodes if their corresponding variables appear in the scope of some function. To obtain a primal graph of an ID, the information arcs and utility nodes should be removed after moralization.

**Mini-bucket Tree Decomposition** The mini-bucket scheme (Dechter and Rish 2003) relaxes an exact tree decomposition (Dechter 2013) by duplicating variables whenever the maximum clique size exceeds an $i$-bound parameter. By splitting a bucket into mini-buckets, the space and time complexity is exponential in the $i$-bound only. The weighted mini-bucket elimination scheme (Liu and Ihler 2011) tightens the mini-bucket relaxation by using Hölder's inequality,

$$\sum_X^{w_X} \otimes_{\alpha=1}^q \Psi_\alpha(\mathbf{X}_\alpha) \leqslant \bigotimes_{\alpha=1}^q \sum_{X_\alpha}^{w_X^\alpha} \Psi_\alpha(\mathbf{X}_\alpha), \qquad (2)$$

where $q$ is the number of mini-buckets generated from the bucket of variable $X$, $\Psi_\alpha(\mathbf{X}_\alpha)$ is the valuation at the $\alpha$-th mini-bucket, $w_X$ is the weight of the variable $X$ that is either 1 for the sum-marginal and 0 for the max-marginal, $w_X^\alpha$ is the set of non-negative weights distributed to $q$ mini-buckets such that $w_X = \sum_{\alpha \in \mathcal{I}_\mathbf{F}} w_X^\alpha$. The weighted mini-buckets reduces back to the naive mini-buckets by assigning one of the weights to 1.0 (sum-marginal) and all others 0.0 (max-marginal). Figure 2 shows the schematic trace of the weighted mini-bucket elimination algorithm for the ID in Figure 1. We can see that the bucket labelled by the variable $S_2$ decomposed into two mini-buckets by duplicating the variable $S_2$ to $S_2'$ and distributing the non-negative weights to $w_{S_2}$ and $w_{S_2'}$ such that $w_{S_2} + w_{S_2'} = 1$. The message from each mini-bucket propagates to the closest bucket labelled

by the variable that appears in the scope of the message after marginalizing the variable $S_2$ and $S_2'$, respectively.

**Join-Graph Decomposition** A join-tree decomposition of a primal graph $\mathcal{G}_p$ is a tree of cliques generated by triangulating the $\mathcal{G}_p$ along a constrained ordering $\mathcal{O}$ compatible with the information constraints of the ID. The space and time complexity of exact inference algorithms for solving Eq. (1) is exponential in the graph parameter called *treewidth* (Dechter 1999). Join-graph decomposition (Mateescu et al. 2010) is an approximation scheme that decomposes the cliques in a tree decomposition, yielding clusters whose scope size is bounded by the $i$-bound, yielding a loopy graph of finer grained clusters. A node in a join-graph is associated with a set of functions and a subset of variables containing their scopes. An edge between two adjacent nodes is labelled by a separator set that is a subset of variables shared between the two nodes. A valid join-graph must satisfy the running intersection property; for each variable $X \in \mathbf{X}$, the set of clusters containing variable $X$ in their scopes induces a connected subgraph. Such a join-graph can be constructed by connecting the mini-buckets in a mini-bucket tree by a chain whose separator is the single variable of the bucket. For example, the mini-bucket tree in Figure 2 can be turned into a join graph by connecting the two mini-buckets for the variable $S_2$ with a separator set $\{S_2\}$.

The generalized dual decomposition scheme (Ping, Liu, and Ihler 2015) (GDD) provides upper bounds to the marginal MAP query by generalizing the Hölder's inequality in Eq. (2) to the fully decomposed setting expressed by

$$\sum_\mathbf{X}^\mathbf{w} \otimes_{\alpha \in \mathcal{I}_\alpha} \Psi_\alpha(\mathbf{X}_\alpha) \leqslant \bigotimes_{\alpha \in \mathcal{I}_\alpha} \sum_{\mathbf{X}_\alpha}^{\mathbf{w}^\alpha} \Psi_\alpha(\mathbf{X}_\alpha), \qquad (3)$$

where $\mathcal{I}_\alpha$ is the index set of all nodes in a join graph, $\Psi_\alpha(\mathbf{X}_\alpha)$ is a valuation that combines all valuations in the $\alpha$-th cluster, $\mathbf{w} = \{w_{X_1}, \cdots, w_{X_{|\mathbf{X}|}}\}$ is the set of all weights corresponds to the set of all variables $\mathbf{X}$, $\mathbf{X}_\alpha$ is the set of duplicated copies of all variables to the $\alpha$-th cluster, and $\mathbf{w}^\alpha = \{w_{X_1}^\alpha, \cdots, w_{X_{|\mathbf{X}|}}^\alpha\}$ is the set of weights to the $\mathbf{X}_\alpha$ such that $w_{X_i} = \sum_{\alpha \in \mathcal{I}_\alpha} w_{X_i}^\alpha$ for all $X_i \in \mathbf{X}$.

**Reparameterized Decomposition Bounds** The upper bounds provided by the various decomposition schemes can be tightened by introducing auxiliary optimization parameters to the decomposition bounds, resulting in reparameterizing of the original functions.

Most recently, (Lee, Ihler, and Dechter 2018) presented a GDD reparameterized bounds for MEU task by extending the fully decomposed bounds over a join-graph decomposition of IDs. The auxiliary optimization parameters are the cost shifting valuations $\delta_{(\mathcal{C}_i, \mathcal{C}_j)}$ and $\delta_{(\mathcal{C}_j, \mathcal{C}_i)}$ between two nodes $\mathcal{C}_i$ and $\mathcal{C}_j$ defined over the separator variables $\mathcal{S}_{(\mathcal{C}_i, \mathcal{C}_j)}$ such that both cancels to the identity $\delta_{(\mathcal{C}_i, \mathcal{C}_j)} \otimes \delta_{(\mathcal{C}_j, \mathcal{C}_i)} = (1, 0)$, and the weight parameters $\mathbf{w}^\mathcal{C}$ that are distributed to each cluster $\mathcal{C}$. From Eq. (3), we can rewrite the reparameterized bound for IDs as,

$$\sum_\mathbf{X}^\mathbf{w} \bigotimes_{\alpha \in \mathcal{I}_\alpha} \Psi_\alpha(\mathbf{X}_\alpha) \leqslant \bigotimes_{\alpha \in \mathcal{I}_\alpha} \sum_{\mathbf{X}_\alpha}^{\mathbf{w}^\alpha} [\Psi_\alpha(\mathbf{X}_\alpha) \otimes (\bigotimes_{(\alpha, \mathcal{C}_j) \in \mathcal{S}} \delta_{(\alpha, \mathcal{C}_j)})], \qquad (4)$$

where each valuation $\Psi_\alpha$ at the $\alpha$-th cluster is reparameterized by the costs from all adjacent edges in the join graph. The local optimum that tightens the MEU can be obtained by minimizing the value component of the right-hand side of Eq. (4) relative to the optimization variables $\mathbf{w}^\alpha$ for all $\alpha \in \mathcal{I}_\alpha$, and $\delta_{(\mathcal{C}_i,\mathcal{C}_j)}$ and $\delta_{(\mathcal{C}_j,\mathcal{C}_i)}$ for all $(\mathcal{C}_i,\mathcal{C}_j) \in \mathcal{S}$, subject to the constraints $w_X = \sum_\alpha w_X^\alpha$ for all $X \in \mathbf{X}$, and $\delta_{(\mathcal{C}_i,\mathcal{C}_j)} \otimes \delta_{(\mathcal{C}_j,\mathcal{C}_i)} = (1,0)$ for all $(\mathcal{C}_i,\mathcal{C}_j) \in \mathcal{S}$.

## Weighted Mini-Bucket Bounds for IDs

The value component of the decomposition bounds for IDs as an Eq. (4) does not have a convex form because the global expected utility value combines the probability and value components from all the decomposed clusters. This non-convexity degrades the quality of the upper bounds computed by algorithms that optimize the bounds. For example, the JGDID scheme (Lee, Ihler, and Dechter 2018) often shows degradation of the quality of the upper bounds even with a higher $i$-bound; the number of optimization parameters is exponential in the $i$-bound, so the dimension of the parameter space rapidly increases.

An alternative approach we explore here is to interleave the variable elimination and decomposition/optimization of the clusters on-the-fly while performing a variable elimination scheme. In this way, the intermediate reparameterization step optimizes a partial decomposition scheme applied to a single cluster of the join-tree only, resulting in a lower dimensional optimization space. In the following subsections, we develop the weighted mini-bucket elimination bounds for IDs (WMBE-ID) based on this idea.

### Derivation of the Bounds

Given an ID $\mathcal{M} := \langle \mathbf{X}, \mathbf{\Psi}, \mathcal{O} \rangle$, we apply the weighted mini-bucket decomposition by Eq. (2) for one variable at a time following the constrained elimination ordering $\mathcal{O} : \{X_{|\mathbf{X}|}, X_{|\mathbf{X}|-1}, \cdots, X_1\}$. The intermediate messages are sent to lower mini-buckets as illustrated in Figure 2. To tighten the upper bound, the auxiliary valuations between mini-buckets can be introduced, yielding the following parameterized bound for each bucket independently,

$$\sum_X^{w_X} \bigotimes_{\alpha=1}^q \Psi_\alpha(\mathbf{X}_\alpha) \leqslant \bigotimes_{\alpha=1}^q \sum_{X_\alpha}^{w_X^\alpha} [\Psi_\alpha(\mathbf{X}_\alpha) \otimes \frac{\delta_{(\alpha-1,\alpha)}(X)}{\delta_{(\alpha,\alpha+1)}(X)}], \quad (5)$$

where $\delta_{(\alpha,\alpha+1)}(X)$ is the cost shifting valuation between mini-buckets from the $\alpha$-th mini-bucket to the $(\alpha+1)$-th mini-bucket. Following the example in Figure 2, the parameterized upper bound to the weighted mini-bucket decomposition at Bucket $S_2$ can be written as,

$$[\sum_{S_2}^{w_{S_2}} \frac{(\lambda^{D_1}, \eta^{D_1})}{\delta_{(S_2,S_2')}}] \otimes [\sum_{S_2'}^{w_{S_2'}} (P(S_2'|S_0,S_1),0) \otimes \delta_{(S_2,S_2')}].$$

However, the value component of the parameterized bound in Eq (5) cannot be served as an objective function for tightening the upper bound to the MEU because the scope size of the combined valuations from mini-buckets after eliminating variable $X_\alpha$ at the right-hand side of the inequality is as large

as the induced width. Therefore, we propose the following surrogate objective function for minimizing the upper bound as follows.

**Theorem 1.** *(weighted MBE Bounds for IDs)* Given an ID $\mathcal{M} := \langle \mathbf{X}, \mathbf{\Psi}, \mathcal{O} \rangle$ *and a constrained variable elimination ordering* $\mathcal{O} := \{X_{|\mathbf{X}|}, X_{|\mathbf{X}|-1}, \cdots, X_1\}$, *assume that the variables* $\{X_{|\mathbf{X}|}, X_{|\mathbf{X}|-1}, \cdots X_{n+1}\}$ *are already eliminated by weighted mini-bucket elimination algorithm. Let* $\Psi^{X_i}(\mathbf{X}_{1:i})$ *be the combination of the valuations allocated to bucket* $X_i$ *of the join-tree,* $Q_{X_i} := \{1, \cdots, q_{X_i}\}$ *be the mini-bucket partitioning for bucket* $X_i$, *and* $\Psi_\alpha^{X_n}(\mathbf{X}_\alpha^{X_n})$ *be the combination of the valuations allocated at the* $\alpha$-th *mini-bucket. Then, the exact MEU of the subproblem defined over variables* $\mathbf{X}_{1:n} := \{X_1, \cdots, X_n\}$ *can be bounded by*

$$\sum_{\mathbf{X}_{1:n}}^{\mathbf{w}_{1:n}} [\bigotimes_{i=1}^n \Psi^{X_i}(\mathbf{X}_{1:i})] \tag{6}$$

$$\leqslant \sum_{\mathbf{X}_{1:n-1}}^{\mathbf{w}_{1:n-1}} [\bigotimes_{i=1}^{n-1} \Psi^{X_i}(\mathbf{X}_{1:i})] \otimes [\bigotimes_{\alpha \in Q_{X_n}} \sum_{X_n}^{w_{X_n}^\alpha} \Psi_\alpha^{X_n}(\mathbf{X}_\alpha^{X_n})] \tag{7}$$

$$\leqslant \bigotimes_{i=1}^n [\bigotimes_{\alpha \in Q_{X_i}} \sum_{\mathbf{X}_{1:n}}^{\mathbf{w}_{1:n}^{X_i,\alpha}} \Psi_\alpha^{X_i}(\mathbf{X}_\alpha^{X_i})] \tag{8}$$

*The weights* $\mathbf{w}_{1:n} := \{w_{X_1}, \cdots w_{X_n}\}$ *in Eq. (6) is the set of weights of the variables* $\mathbf{X}_{1:n}$, *each of them is either 1 for* $X_i$ *being a chance variable or 0 for a decision variable, and the weights* $\mathbf{w}_{1:n}^{X_i,\alpha}$ *in Eq. (8) is the set of weights of the variables* $\mathbf{X}_{1:n}$ *in the* $\alpha$-th *mini-bucket partition in bucket* $X_n$ *such that* $w_{X_k} = \sum_{i=1}^n \sum_{\alpha \in Q_{X_i}} w_{X_k}^{X_i,\alpha}$.

*Proof.* The upper bound of Eq. (7) can be obtained by applying the weighted mini-bucket scheme in Eq. (2) to the bucket $X_n$, and the upper bound of Eq. (8) can be obtained by first partitioning all buckets to mini-buckets and applying the fully decomposed bound of Eq. (3) to each mini-bucket, $\alpha \in Q_{X_i}$ for all $X_i \in \mathbf{X}$ (Ping, Liu, and Ihler 2015; Lee, Ihler, and Dechter 2018). $\square$

### Optimizing the Upper Bounds

**Optimization Objectives and Parameters** The upper bound derived in Theorem 1 can be reparameterized by the cost functions on the chain of mini-buckets before processing and sending messages. Given an ID $\mathcal{M} := \langle \mathbf{X}, \mathbf{\Psi}, \mathcal{O} \rangle$ and a constrained variable elimination ordering $\mathcal{O} := \{X_{|\mathbf{X}|}, X_{|\mathbf{X}|-1}, \cdots, X_1\}$, the weighted mini-bucket bounds for IDs in Theorem 1 can be parameterized over the chain of mini-buckets $Q_{X_n}$ as follows, assuming that variable $X_n$ is removed after reparameterization.

$$[\bigotimes_{i=1}^{n-1} \bigotimes_{\alpha \in Q_{X_i}} \sum_{\mathbf{X}_{1:n-1}}^{\mathbf{w}_{1:n-1}^{X_i,\alpha}} \Psi_\alpha^{X_i}(\mathbf{X}_\alpha^{X_i})] \otimes$$

$$[\bigotimes_{\alpha \in Q_{X_n}} \sum_{\mathbf{X}_{1:n}}^{\mathbf{w}_{1:n}^{X_n,\alpha}} \Psi_\alpha^{X_n}(\mathbf{X}_\alpha^{X_n}) \frac{\delta_{(\alpha-1,\alpha)}^{X_n}(X_n)}{\delta_{(\alpha,\alpha+1)}^{X_n}(X_n)}] \tag{9}$$

**Algorithm 1** Weighted Mini-Bucket Elimination Bounds for IDs (WMBE-ID)

---

**Require:** Influence diagram $\mathcal{M} = \langle \mathbf{X}, \boldsymbol{\Psi}, \mathcal{O} \rangle$, total constrained elimination order $\mathcal{O} := \{X_N, X_{N-1}, \cdots, X_1\}$, $i$-bound, iteration limit $L$,

**Ensure:** an upper bound of the MEU

1: Initialization: Generate a schematic mini-bucket tree (Dechter and Rish 2003) and allocate valuations to proper mini-buckets.
2: $Ub \leftarrow (1, 0)$
3: **for** $i \leftarrow N$ to 1 **do**
4:    Partition bucket $X_i$ to mini-buckets $Q_{X_i} := \{1, \cdots, q_{X_i}\}$ with $i$-bound
5:    **for** $\alpha \in Q_{X_i}$ **do**
6:      $\Psi_\alpha^{X_i}(\mathbf{X}_\alpha^{X_i}) \leftarrow$ combine valuations at the mini-bucket $\alpha$
7:    **end for**
8:    $iter = 0$
9:    Initialize join-graph with the uniform weights for all remaining mini-buckets $\{Q_{X_1}, \cdots Q_{X_n}\}$
10:    Evaluate objective function Eq. (9) for all remaining mini-buckets $\{Q_{X_1}, \cdots Q_{X_n}\}$
11:    **while** $iter < L$ or until bounds improved **do**
12:      Update a set of cost functions $\{\delta_{(\alpha, \alpha+1)|\alpha \in Q_{X_i}}\}$ subject to the constraints in Eq. (13) and (14)
13:      Update a set of weights $\{w_{X_i}^{X_i, \alpha}|\alpha \in Q_{X_i}\}$
14:    **end while**
15:    **for** $\alpha \in Q_{X_i}$ **do**
16:      $(\lambda_\alpha^{X_i}, \eta_\alpha^{X_i}) \leftarrow \sum_{X_i}^{w_{X_i}^{X_i, \alpha}} \tilde{\Psi}_\alpha^{X_i}(\mathbf{X}_\alpha^{X_i})$
17:      **if** $(\lambda_\alpha^{X_i}, \eta_\alpha^{X_i})$ is constant **then**
18:        $Ub \leftarrow Ub \otimes (\lambda_\alpha^{X_i}, \eta_\alpha^{X_i})$
19:      **else**
20:        Send message $(\lambda_\alpha^{X_i}, \eta_\alpha^{X_i})$ to the closest bucket labelled by the variable appearing in the scope of the message
21:      **end if**
22:    **end for**
23: **end for**
24: Return value component of $Ub$

---

The optimization parameters are the set of cost functions between two mini-buckets and the weights over the mini-buckets $Q_{X_n}$,

$$\{\delta_{(\alpha, \alpha+1)}^{X_n}(X_n)|\forall \alpha, \alpha+1 \in Q_{X_n}\} \quad (10)$$

$$\{w_{X_n}^{X_n, \alpha}|\forall \alpha, \alpha+1 \in Q_{X_n}\}, \quad (11)$$

where $\delta_{(\alpha, \alpha+1)}^{X_n}(X_n)$ is defined by the probability component $\lambda_{(\alpha, \alpha+1)}^{X_n}(X_n)$ and the value component $\eta_{(\alpha, \alpha+1)}^{X_n}(X_n)$. The objective function for the optimization can be defined by the value component of Eq. (9),

$$\sum_{i=1}^{n-1} \sum_{\alpha \in Q_{X_i}} \frac{\sum_{\mathbf{X}_{1:n-1}}^{\mathbf{w}_{1:n-1}^{X_i, \alpha}} V_\alpha^{X_i}}{\sum_{\mathbf{X}_{1:n-1}}^{\mathbf{w}_{1:n-1}^{X_i, \alpha}} P_\alpha^{X_i}} + \quad (12)$$

$$\sum_{\alpha \in Q_{X_n}} \frac{\sum_{\mathbf{X}_{1:n}}^{\mathbf{w}_{1:n}^{X_n, \alpha}} P_\alpha^{X_n} \frac{\lambda_{(\alpha-1, \alpha)}^{X_n}}{\lambda_{(\alpha, \alpha+1)}^{X_n}} \left[ \frac{V_\alpha^{X_n}}{P_\alpha^{X_n}} - \frac{\eta_{(\alpha, \alpha+1)}^{X_n}}{\lambda_{(\alpha, \alpha+1)}^{X_n}} + \frac{\eta_{(\alpha-1, \alpha)}^{X_n}}{\lambda_{(\alpha-1, \alpha)}^{X_n}} \right]}{\sum_{\mathbf{X}_{1:n}}^{\mathbf{w}_{1:n}^{X_n, \alpha}} P_\alpha^{X_n} \frac{\lambda_{(\alpha-1, \alpha)}^{X_n}}{\lambda_{(\alpha, \alpha+1)}^{X_n}}},$$

where $\Psi_\alpha^{X_i} := (P_\alpha^{X_i}, V_\alpha^{X_i})$ after omitting the scope $\mathbf{X}_\alpha^{X_i}$. Note that, the evaluation of upper bounds while performing

the weighted mini-bucket elimination requires reconfiguring the join-graph and weight parameters of the remaining mini-buckets after eliminating a variable. However, the evaluation of the optimization objective inside optimization procedure does not require re-evaluation of all mini-buckets because the fully decomposed bounds of all nodes except the mini-buckets under optimization does not change subject to the changes in the cost functions and weights. In the empirical evaluation, the cost functions and weights are updated separately by calling the constrained optimization routines to update the cost and the exponentiated gradient descent (Kivinen and Warmuth 1997) for updating the weights of mini-buckets.

**Constraints for Optimizing Cost Functions** Since the powered-sum elimination operator is defines over the absolute value of a function, the value components at all mini-buckets are constrained to remain positive after reparameterization. Let $\Psi_\alpha^{X_n}(\mathbf{X}_\alpha)$ be the valuation at the $\alpha$-th mini-bucket of bucket $X_n$ with the probability component $\lambda_\alpha^{X_n}(\mathbf{X}_\alpha)$ and the value component $\eta_\alpha^{X_n}(\mathbf{X}_\alpha)$, and $\delta_{(\alpha, \alpha+1)}(X_n) := (\lambda_{(\alpha, \alpha+1)}(X_n), \eta_{(\alpha, \alpha+1)}(X_n))$ be the cost shifting valuation between the $\alpha$-th and $(\alpha+1)$-th mini-buckets. Then, the non-negativity of the value components after the reparameterization is ensured by:

$$\frac{\eta_\alpha^{X_n}(\mathbf{X}_\alpha)}{\lambda_\alpha^{X_n}(\mathbf{X}_\alpha)} - \frac{\eta_{(\alpha, \alpha+1)}(X_n)}{\lambda_{(\alpha, \alpha+1)}(X_n)} + \frac{\eta_{(\alpha-1, \alpha)}(X_n)}{\lambda_{(\alpha-1, \alpha)}(X_n)} \geqslant 0. \quad (13)$$

In addition, the non-negativity of probability components is ensured by:

$$\lambda_{(\alpha, \alpha+1)}(X_n) \geqslant 0 \quad (14)$$

for all mini-buckets $\alpha \in Q_{X_n}$. Equipped with the optimization objective and constraints, any constrained optimization procedure can be applied to reparameterize the mini-buckets. For the empirical evaluation, we integrated off-the-shelf optimization libraries such as sequential least square programming (Kraft 1988).

**Interleaving Elimination and Optimization** Algorithm 1 outlines the overall procedure of the weighted mini-bucket elimination interleaved with reparameterization to compute the upper bound of MEU. Given an input ID $\mathcal{M}$ and a total constrained elimination order $\mathcal{O}$, the schematic bucket tree elimination algorithm (Dechter 1999) is called to generate a join-tree and allocate valuations at the initialization step (line 1). Variables are processed from first to last in the ordering, as follows. Given the current variable $X_i$, the algorithm partitions its bucket into mini-buckets $Q_{X_i}$ and combines the valuations placed in each mini-bucket (lines 4–7). The fully decomposed join-graph decomposition based bound is pre-computed using the uniform weights at all mini-buckets remaining in the problem (lines 9–10). Subsequently, the cost functions and weights that parameterize the mini-buckets corresponding to variable $X_i$ are updated in order to tighten the upper bound of the inequality Eq. (9). After the optimization step, messages from mini-buckets are computed by marginalizing the reparameterized valuations $\tilde{\Psi}_\alpha^{X_i}$ using the powered-sum operator with weights $\{w_{X_i}^{X_i, \alpha}|\forall \alpha \in Q_{X_i}\}$.

| Domain | $n$ | $f$ | $k$ | $s$ | $w$ |
|---|---|---|---|---|---|
| FH-MDP | 99, 145, 170 | 120, 170, 240 | 3, 3, 5 | 7, 9, 9 | 21, 39, 43 |
| FH-POMDP | 57, 92, 96 | 72, 128, 140 | 2, 2, 3 | 5, 6, 9 | 28, 43, 47 |
| RAND | 60, 77, 91 | 60, 77, 91 | 2, 2, 2 | 3, 3, 3 | 20, 27, 41 |
| BN | 54, 54, 100 | 54, 54, 100 | 2, 2, 2 | 6, 10, 10 | 19, 24, 28 |

Table 1: Benchmark statistics. We show the minimum, median, and maximum values for each of the problem parameters: $n$ – the number of chance and decision variables, $f$ – the number of probability and utility functions, $k$ – the domain size, $s$ – the scope size, and $w$ – the induced width, respectively.

## Experiments

We compare the performance of our proposed bounding scheme WMBE-ID with earlier approaches on 4 domains each containing 5 problem instances. The benchmark statistics are summarized in Table 1.

### Benchmarks

For our purpose, we generated 4 domains in the following way: (1) Factored FH-MDP instances are generated from two stage factored MDP templates by varying the number of state and action variables, the scope size of functions, and the length of time steps between 3 and 10. (2) Factored FH-POMDP instances are generated similarly to MDP instances, but it incorporates observed variables. (3) Random influence diagrams (RAND) are generated from a random topology of influence diagram by varying the number of chance, decision, and value nodes. (4) BN instances are IDs converted from the Bayesian network released in the UAI-2006 probabilistic inference challenge by converting random nodes to decision nodes and adding utility nodes.

### Algorithms

We evaluate the proposed WMBE-ID algorithm in 3 different configurations: (1) uniform weights without cost updates (WMBE-U) (2) uniform weights with cost updates (WMBE-UC), and (3) update both weights and costs (WMBE-WC). For comparison, we consider the following earlier approaches: the mini-bucket elimination bound (MBE), MBE combined with the re-ordering relaxation (MBE-Re), and the state-of-the-art join graph decomposition bounds for IDs (JGDID). We implemented all algorithms in Python using the NumPy (Oliphant 2015) and SciPy (Jones et al. 2001 ) libraries. WMBE-U, MBE, and MBE-Re are non-iterative algorithm that computes the upper bounds in a single pass. On the other hand, WMBE-UC, WMBE-WC, and JGDID are iterative algorithms that reparameterize cost functions and weights until the iteration limit or convergence. The number of iterations of WMBE-UC and WBME-WC is the maximum number of calls allowed to reparameterizing the cost functions of mini-buckets by the off-the-shelf optimization library, sequential least square programming in SciPy (Jones et al. 2001 ) with the default parameters. JGDID updates all parameters by the gradient descent method at each iteration.

### Comparing on Individual Instances

Table 2 shows the quality of upper bounds of all 6 algorithms at each instance of four domains with the $i$-bounds 1, 5, 10, and 15, and the iteration limit 1, 5, 10, and 20. We can see that the quality of the bound from MBE and MBE-Re is a magnitude worse than the other algorithms. JGDID algorithm generates the most tight bound on many of the cases but it consistently produces worse bounds with higher $i$-bound and it takes more time. On the other hand, WMBE-ID algorithms consistently improve the quality of the bounds with higher $i$-bounds.

Comparing the 3 variants of WMBE-ID algorithms, optimizing both weights and cost functions greatly improved the quality of bound with additional time overhead. In case of `ID from BN 78 w19d3` we can also observe that WMBE-WC produces better bound than the best bound produced by JGDID; 23.44 in 1078 seconds by WMBE-WC with $i$-bound 15, and 27.53 in 1281 seconds by JGDID with $i$-bound 1. Similarly, WBME-WC produced better bounds for `mdp9-32-3-8-3` instance; 21.48 in 5905 seconds by WMBE-WC with $i$-bound 10, and 23.58 in 15340 seconds by JGDID with $i$-bound 1.

### Comparing WMBE-ID vs. JGDID

Table 3 compares the quality of the upper bounds as well as the running time against JGDID($i$=1). Clearly, we can see that JGDID with $i$-bound 10 shows degradation of the quality of the bounds on all 20 instances. On the other hand, both WMBE-UC and WMBE-WC improves the upper bounds with higher $i$-bounds, and WMBE-WC produces tighter bounds than JGDID on 7 instances with the $i$-bound 15 in shorter time bounds than JGDID($i$=1).

## On Using WMBE-ID as Heuristics

The experiments shows that WMBE-ID produces high quality bounds in a shorter time bounds compared to JGDID, and it improves the tightness of the bounds with higher $i$-bounds as opposed to JGDID. More importantly, WMBE-ID can be pre-compiled as a static heuristic function if all the intermediate messages are stored as a look up table before starting search. This characteristic is especially desired for heuristic evaluation functions that require less overhead on computing the heuristic values. The weighted mini-bucket heuristic functions for MAP and MMAP have shown state-of-the-art performance when used to guide AND/OR search strategies (Otten and Dechter 2012; Marinescu et al. 2018). Therefore our plan is to integrate WMBE-ID as a heuristic generator for AND/OR search algorithms for solving IDs.

## Conclusion

We presented a new bounding scheme for influence diagrams, called WMBE-ID, which computes upper bounds of the MEU by interleaving variable elimination with optimizing partial decomposition within each variable's bucket. Compared with the previous approaches, our proposed upper bounding scheme produces high quality upper bounds in shorter time bounds. This is instrumental for our plan to

| Instance (n,f,k,s,w) | Algorithm | i=1 | | | | i=5 | | | |
|---|---|---|---|---|---|---|---|---|---|
| | | iter=1 | 5 | 10 | 20 | iter=1 | 5 | 10 | 20 |
| ID_from_BN_78_w19d3 (54, 54, 2, 10, 19) | WMBE-U | (47, 418) | | | | (52, 147.78) | - | - | - |
| | WMBE-UC | (277, 381) | (532, 381) | (430, 381) | (484, 381) | (236, 151.58) | (378, 151,58) | (488, 151.58) | (400, 151.58) |
| | WMBE-WC | (30, 237) | (660, 235) | (1089, 228) | (803, 235) | (292, 77.75) | (709, 84.18) | (625, 61.35) | (770, 77.32) |
| | JGDID | (0.51, 889) | (973, 28.34) | **(1281, 27.53)** | - | (0.7, 4362) | (915, 118) | (2731, 33.64) | (6076, 32.59) |
| | MBE | (1, 314234) | - | - | - | (1, 2957) | - | - | - |
| | MBE-Re | (1, 245894) | - | - | - | (1, 19059) | - | - | - |

| | Algorithm | i=10 | | | | i=15 | | | |
|---|---|---|---|---|---|---|---|---|---|
| | | iter=1 | 5 | 10 | 20 | iter=1 | 5 | 10 | 20 |
| | WMBE-U | (214, 46.93) | - | (539, 44.94) | (467, 44.94) | (252, 27,79) | - | (586, 27.79) | (259, 27.79) |
| | WMBE-UC | (341, 44.95) | (489, 44.94) | | | (247, 27,79) | (270, 27.79) | | |
| | WMBE-WC | (291, 38.20) | (501, 38.34) | (557, 37.84) | (557, 37.84) | (333, 23,56) | (438, 23.44) | (715, 23.44) | **(1078, 23.44)** |
| | JGDID | (1, 4561) | (3530, 90.23) | (7433, 48.15) | (15235, 47.08) | (1, 4513) | (7391, 45.21) | (15720, 42.34) | - |
| | MBE | (1, 113) | - | - | - | (1, 37.38) | - | - | - |
| | MBE-Re | (1, 329) | - | - | - | (1, 49.43) | - | - | - |

| Instance (n,f,k,s,w) | Algorithm | i=1 | | | | i=5 | | | |
|---|---|---|---|---|---|---|---|---|---|
| | | iter=1 | 5 | 10 | 20 | iter=1 | 5 | 10 | 20 |
| mdp9-32_3_8_3 (99, 120, 3, 9, 43) | WMBE-U | (466, 2.86E+9) | - | - | - | (494, 1.25E+8) | - | - | - |
| | WMBE-UC | (6040, 1.01E+9) | (8052, 1.91E+9) | (9991, 1.00E+9) | (7866, 1.00E+9) | (2146, 7.48E+7) | (4169, 7.48E+7) | (5667, 7.48E+7) | (6718, 7.48E+7) |
| | WMBE-WC | (8017, 30.6) | (21496, 69.53) | (10085, 2628) | (17214, 80.78) | (6691, 23.89) | (16113, 116.13) | (14653, 357) | (16888, 59.16) |
| | JGDID | (2.8, 1.65E+11) | (7625, 48.22) | **(15340, 23.58)** | - | (3, 2.58E+13) | (6790, 7104) | (18589, 634) | (30243, 27.57) |
| | MBE | (3, 2.4E+21) | - | - | - | (3, 4.3E+14) | - | - | - |
| | MBE-Re | - | - | - | - | - | - | - | - |

| | Algorithm | i=10 | | | | i=15 | | | |
|---|---|---|---|---|---|---|---|---|---|
| | | iter=1 | 5 | 10 | 20 | iter=1 | 5 | 10 | 20 |
| | WMBE-U | (581, 1666593) | - | - | - | (790, 372057) | - | - | - |
| | WMBE-UC | (3047, 825028) | (3720, 825513) | (4747, 824649) | (6327, 824641) | (4054, 319426) | (6330, 33415) | (7109, 321955) | (5019, 321183) |
| | WMBE-WC | **(5905, 21.48)** | (7588, 146) | (11945, 105) | (143243,. 46.45) | (8266, 20,47) | (14521, 78.54) | (20770, 69.13) | (24150, 45.5) |
| | JGDID | (4, 4.01E+14) | (4282, 8.75E+8) | (9294, 4.40E+7) | (21315, 4.62E+6) | (7, 4.63E+14) | (9250, 2.42E+8) | (19791, 5.35E+7) | (40074, 7.39E+5) |
| | MBE | (2.6, 159E+12) | - | - | - | (2, 1.73E+10) | - | - | - |
| | MBE-Re | - | - | - | - | - | - | - | - |

| Instance (n,f,k,s,w) | Algorithm | i=1 | | | | i=5 | | | |
|---|---|---|---|---|---|---|---|---|---|
| | | iter=1 | 5 | 10 | 20 | iter=1 | 5 | 10 | 20 |
| pomdp8-14_9_3_12_4 (96, 140, 2, 6, 47) | WMBE-U | (312, 9.3E+14) | - | - | - | (2, 3.4E+14) | - | - | - |
| | WMBE-UC | (731, 9.3E+14) | (2105, 9.3E+14) | (5674, 9.3E+14) | (8311, 9.3E+14) | (2744, 2.4E+13) | (2616, 2.4E+13) | (2815, 2.4E+13) | (2815, 2.4E+13) |
| | WMBE-WC | (672, 1.8E+13) | (2640, 1.27E+13) | (4434, 1.24E+13) | (8105, 1.24E+13) | (549, 1E+12) | (2747, 7.23E+11) | (3516, 7.23E+11) | (4770, 7.21E+11) |
| | JGDID | (2.05, 1.8E+16) | (1920, 3.73E+10) | (4458, 8.65E+9) | **(7111, 6.12E+8)** | (3, 2.75E+19) | - | - | - |
| | MBE | (3, 1.94E+22) | - | - | - | (3, 2.57E+18) | - | - | - |
| | MBE-Re | (4, 2.9E+18) | - | - | - | (2, 3.4E+14) | - | - | - |

| | Algorithm | i=10 | | | | i=15 | | | |
|---|---|---|---|---|---|---|---|---|---|
| | | iter=1 | 5 | 10 | 20 | iter=1 | 5 | 10 | 20 |
| | WMBE-U | (755, 1.90E+10) | - | - | - | (1861, 9.49E+8) | - | - | - |
| | WMBE-UC | (832, 1.90E+10) | (2145, 1.90E+10) | (3391, 1.90E+10) | (3057, 1.90E+10) | (35236, 1.45E+7) | (55864, 1.45E+7) | (61070, 14503836) | (61401, 1.45E+7) |
| | WMBE-WC | (904, 4866832237( | (1532, 4.95E+9) | (3301, 4.92E+9) | (5855, 4.88E+9) | (35313, 2.87E+7) | (67977, 8.97E+6) | (81550, 8.97E+6) | (87807, 1.45E+11) |
| | JGDID | (4, 3.2E+18) | (3224, 3.48E+12) | (7101, 5.55E+11) | (14635, 4.42E+9) | (12, 1.4E+18) | (12196, 4.39E+11) | (28338, 1.29E+11) | (49188, 6.81E+9) |
| | MBE | (2, 1.4E+15) | - | - | - | (3, 7.3E+12) | - | - | - |
| | MBE-Re | (1, 326947477) | - | - | - | (1, 284607) | - | - | - |

| Instance (n,f,k,s,w) | Algorithm | i=1 | | | | i=5 | | | |
|---|---|---|---|---|---|---|---|---|---|
| | | iter=1 | 5 | 10 | 20 | iter=1 | 5 | 10 | 20 |
| rand-c70d7o1-01 (91, 91, 2, 3, 41) | WMBE-U | (64, 1028113) | - | - | - | (109, 9227) | - | - | - |
| | WMBE-UC | (183, 1.04E+6) | (327, 1.04E+6) | (489, 1.04E+6) | (620, 1.04E+6) | (329, 7588) | (346, 8196) | (732, 8196) | (613, 7625) |
| | WMBE-WC | (287, 316431) | (779, 413886) | (655, 305882) | (820, 293482) | (354, 3782) | (461, 4021) | (1351, 3886) | (2649, 4092) |
| | JGDID | (1, 2.15E+7) | (337, 61104) | (842, 758) | **(1453, 686)** | (1, 1.84E+7) | (1248, 11213) | (3195, 762) | (4397, 736) |
| | MBE | (1, 2.69E+9) | - | - | - | (1, 445863) | - | - | - |
| | MBE-Re | (1, 2.61E+9) | - | - | - | (2, 739433) | - | - | - |

| | Algorithm | i=10 | | | | i=15 | | | |
|---|---|---|---|---|---|---|---|---|---|
| | | iter=1 | 5 | 10 | 20 | iter=1 | 5 | 10 | 20 |
| | WMBE-U | (160, 1863) | - | - | - | (193, 1542) | - | - | - |
| | WMBE-UC | (765, 1835) | (1225, 1820) | (1463, 1819) | (1058, 1820) | (806, 1521) | (1066, 1518) | (1734, 1516) | (1113, 1512) |
| | WMBE-WC | (428, 1812) | (706, 1731) | (418, 1717) | (1234, 1749) | (1028, 1510) | (1638, 1576) | (921, 1576) | (1892, 1576) |
| | JGDID | (1, 20049757) | (4228, 2167) | (8358, 1303) | (15787, 795) | (2, 2.37E+7) | (7837, 2006) | (15134, 1357) | - |
| | MBE | (1, 4937) | - | - | - | (1, 7027) | - | - | - |
| | MBE-Re | (1, 40695) | - | - | - | (1, 16139) | - | - | - |

Table 2: The performance of the bounding schemes on individual instances. n is the number of variables, f is the number of functions, k is the maximum domain size, s is the maximum scope size, w is the constrained induced width. We show the (time, upper bound) for various i-bounds and number of iterations for algorithms updating the costs or weights. WMBE-U is the mini-bucket elimination with uniform weights, WMBE-UC preforms cost shifting without optimizing the weight, WMBE-WC optimizes both weights and costs, JGDID is the fully decomposed bound over a join graph that optimizes both weights and costs, MBE is the simple mini-bucket elimination, and MBE-Re is mini-bucket elimination with relaxed variable ordering. MBE, MBE-RE, and WMBE-U do not optimize the bound. The best upper bounds are highlighted.

| Instance | WMBE-UC | | | | WMBE-WC | | | | JGDID (i=10) | |
|---|---|---|---|---|---|---|---|---|---|---|
| | i=10, iter=5 | | i=15, iter=5 | | i=10, iter=5 | | i=15, iter=5 | | i=10. iter<100 | |
| ID_from_BN_0_w28d6 | 13.89% | 7.42 | 73.66% | 4.34 | 16.56% | 3.36 | 56.41% | 2.93 | 4.10323766 | 1.30 |
| ID_from_BN_0_w29d6 | 15.36% | 1.16E+01 | 18.26% | 3.75 | 19.38% | 5.29 | 19.12% | 2.96 | 2.55257594 | 1.58 |
| ID_from_BN_78_w19d3 | 36.85% | 1.63 | 20.33% | 1.01 | 37.73% | 1.39 | 33.01% | **8.52E-01** | 15.8240436 | 1.70 |
| ID_from_BN_78_w23d6 | 4.53% | 2.59 | 4.91% | 1.40 | 5.38% | 1.33 | 13.45% | 1.10 | 1.27099854 | 1.79 |
| ID_from_BN_78_w24d6 | 11.12% | 5.08 | 12.33% | 2.37 | 8.00% | 4.24 | 22.92% | 1.56 | 1.66577736 | 1.83 |
| mdp5-16_3_8_10 | 5.79% | 1.02E+09 | 38.05% | 5.13E+03 | 11.01% | 8.26 | 49.74% | **7.54E-01** | 1.1375437 | 4.68E+11 |
| mdp6-20_5_5_5 | 14.09% | 2.94E+05 | 20.17% | 5.29E+03 | 48.94% | 6.06 | 27.61% | **8.54E-01** | 3.38715936 | 2.69E+04 |
| mdp7-28_3_6_5 | 10.66% | 4.26E+09 | 13.78% | 1.43E+08 | 18.54% | 4.52 | 43.47% | **9.44E-01** | 1.28478854 | 1.56E+07 |
| mdp8-28_3_6_4 | 15.52% | 2.81E+07 | 23.15% | 5.50E+05 | 38.35% | 5.27E+01 | 46.77% | 9.10 | 2.28022631 | 8.69E+03 |
| mdp9-32_3_8_3 | 13.38% | 3.56E+04 | 22.77% | 1.44E+04 | 27.30% | 6.32 | 52.24% | 3.39 | 1.41892342 | 2.65E+03 |
| pomdp10-12_7_3_8_4 | 25.70% | **5.83E-01** | 246.94% | **1.33E-02** | 28.42% | **2.08E-01** | 282.16% | **8.28E-03** | 2.64238123 | **3.23E-01** |
| pomdp6-12_6_2_6_3 | 53.85% | 1.39E+01 | 26.62% | 7.04 | 29.69% | 6.02 | 52.48% | 5.16E+01 | 1.09018931 | 7.36 |
| pomdp7-20_10_2_10_3 | 22.72% | 6.10E+02 | 46.78% | 5.64 | 27.76% | 5.14E+01 | 44.04% | 4.19 | 1.74323656 | 7.28E+01 |
| pomdp8-14_9_3_12_4 | 19.86% | 3.10E+01 | 45.87% | 1.55 | 14.19% | 8.08 | 60.84% | **5.53E-01** | 1.99441208 | 7.22 |
| pomdp9-14_8_3_10_4 | 36.27% | 2.47E+01 | 503.77% | **1.19E-01** | 39.76% | 6.54 | 666.74% | **5.66E-02** | 3.95469321 | 1.70E+01 |
| rand-c50d15o1-03 | 29.02% | 2.64 | 51.31% | 1.24 | 52.92% | 2.35 | 128.04% | 1.12 | 15.1473433 | 1.35 |
| rand-c50d5o1-01 | 45.44% | **9.11E-01** | 24.30% | **6.75E-01** | 50.52% | **8.46E-01** | 31.86% | **6.82E-01** | 16.8680706 | 1.02 |
| rand-c70d14o1-01 | 45.02% | 2.08E+01 | 470.27% | 8.98 | 59.53% | 8.47 | 319.50% | 9.40 | 18.7158034 | 1.23 |
| rand-c70d21o1-01 | 35.12% | 1.89E+01 | 114.52% | 3.84 | 40.17% | 7.24 | 148.38% | 2.75 | 8.84579305 | 1.16 |
| rand-c70d7o1-01 | 84.32% | 2.65 | 73.37% | 2.21 | 48.63% | 2.52 | 112.71% | 2.30 | 14.7530922 | 1.15 |

Table 3: Comparing the ratio of time and quality of upper bounds against JGDID($i$=1). WMBE-UC and WMBE-WC were provided with $i$-bound 10 and 15 with the number of iteration fixed to 5, and JGDID were provided $i$-bound 1 and 10 with the maximum number iteration limited by 100. All the quantities are normalized by the statistics of JGDID($i$=1).

use such bounds as a heuristic evaluation function for search algorithms for solving influence diagrams.

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
