# OpenReview forum: "A Weighted Mini-Bucket Bound Heuristic for Solving Influence Diagrams"
_icaps-conference.org/ICAPS/2019/Workshop/HSDIP_

### Official Review · AnonReviewer2 · 2019-04-02
**Not sufficiently self-contained for the target audience of HSDIP participants**

**Rating:** 2
**Confidence:** 2

**Review:**

The paper presents an extension for a bucket elimination algorithm to
solve maximum expected utility (MEU) problems. the paper builds
strongly on previous work and is extremely hard to follow without a
background in these methods. I don't have such a background and could
follow the mathematics of the first two pages on an abstract level with
some effort. However, I did not get an intuition on what the introduced
concepts meant and what they were used for. Without an intuition on
the meaning of the notation, it was impossible to dig deeper into the
paper or verify any of the statements. On an abstract mathematical
level, the paper does not contain enough information to verify its
statements which would be fine if the explanations and references would
be sufficient to fill the gaps.

Let me explain my evaluation of a "strong reject" with a low confidence:
Since I see myself as partially representative of the HSDIP audience, I
see this paper as missing its target audience and not being
sufficiently self-contained. Since I cannot verify any of the
statements, I cannot suggest to accept this paper. As long as the
claims are verified by another reviewer (representative for the part of
the HSDIP audience that does have the required background) I'm happy to
let my suggestion to reject be overruled, though. If the other
reviewers had similar problems, though, the paper should not be
accepted without a proper review. In case the paper is accepted, I ask
the authors to make it more accessible to the HSDIP audience and in
particular readers that are not familiar with all previous work.

To make this review more constructive, I'll point out some places that
are hard to understand in the hope that this helps to improve the
paper. While some of them are written as questions, I don't expect any
answers for them in a rebuttal. Instead, I hope that the authors can
use them to see what kind of questions an HSDIP participant might have
about the paper and update the paper so these points are explained in
the paper.

* The Section "Introduction" uses the following terms without
explaining them:
  - MAP and MMAP
  - cost-shifting
  - decomposing
  - mini-bucket tree
  - cluster
  - i-bound
  - partial assignment (that one becomes more clear later on)
  - variable ordering
  - dual decomposition scheme
  - directional
  - information constraints
  - information relaxation scheme
  - chance variables (becomes clear later on but is unclear at that point)
  - consistent variable ordering


* The same section also uses terms that have a reference but since they
are relevant to the paper, they should be explained better instead of
just relying on the reference to clear things up:
  - constrained join tree
  - mini-bucket scheme
  - valuation algebra
  - combination and maginalization
  - potentials

* It might help to move the related work section closer to the end of
the paper when the relevant concepts already have been introduced.

* One very important point to get across to your reader is what solving
an ID means. An example that shows a specific problem modeled as an ID
would be very useful in the introduction or at the start of the
background section. The example in Figure 1 is too generic to explain
this.

* In the section "Background" the following things were unclear to me:
  - In the definition of P, shouldn't it be "i \in I_P" instead of "P_i
  \in I_P"? (same for U)

  - Why does Figure 1 use S instead of C for the chance nodes?

  - How is the set pa(C_i) defined? My guess is that the functions in C
  each have a scope that is relevant here but this is not part of their
  definition.

  - How are the informational arcs defined based on the tuple
  <C, D, P, U, O>? I assume they can be derived from O in some form but
  I don't see how exactly. Also, if this is the case, you should define
  O before defining the informational arcs.

  - What do you mean with "a decision can only be preceded by at most
  one decision"? In the DAG in Figure 1 decision node D_1 is preceded
  by three chance nodes and in the order O, decision D_i is preceded by
  decisions D_0, ..., D_{i-1} and their information sets.

  - In your definition of a history, the node D_i is part of the
  history H(D_i) but you say decision D_i is made based on the history
  H(D_i). Doesn't this mean that D_i is made based on the outcome of
  D_i?

  - What type does D_i have? in the definition of \Delta, it is used
  like a set (as the co-domain of a function).

  - What is the definition of the "relevant history" R(D_i)

  - The valuation algebra works on tuples of functions and defines
  functions change some tuples into other tuples but after reading the
  section I still have no idea what such a tuple represents or why I
  would want to combine them with the defined operations.

  - The type of the tuples in the valuation algebra is also not clear.
  It is introduced as (P(X), V(X)) but then later used as (P_i, 0) and
  (1, U_i). So in the first position, there are numbers (1 and P(X)) or
  functions (P_i) and in the second position, there are numbers (0) or
  functions (U_i). I assume you mean functions in all cases and use 1
  and 0 as constant functions and P(X) as a way to write that a
  function P depends on the variables in set X? If so, being more
  explicit about this would help.

  - In Definition 2 the powered-sum elimination operator depends on X
  and Y but in its definition below, it uses X everywhere. How is it
  defined if Y is different from X?

  - When you write "The MEU can be written as ....", I neither have an
  idea of why this could be true, nor of a way that I could verify
  this.

  - What is a "primal graph" and "moralization"?

---

> ### Author Response · Authors · 2019-04-09
> **Thanks for your review.**
>
> We acknowledge that the submitted paper does not cover the relevant backgrounds on inference over probabilistic graphical models in a concise and complete way to deliver the idea in the paper.
> Since you don't expect any answers for the list of the questions,  we will keep those comments for the future works to make submissions more accessible and clear to the readers. Thanks for the extensive review.

---

> > ### Comment · AnonReviewer2 · 2019-04-09
> > **Example for a conformant planning problem mapped to an influence diagram**
> >
> > Thanks for your response. I think one easy step to make the paper more accessible would be to show a concrete problem represented as an influence diagram. Since you mentioned it in your other response, I would be interested in seeing an example of a conformant planning problem as an influence diagram. Just as an example, there is the BTUC problem, described by Cimatti and Roveri (https://www.jair.org/index.php/jair/article/view/10267/24466) but any other example would be great as well (I just picked the first result from Google scholar).

---

> > > ### Author Response · Authors · 2019-04-10
> > > **Sketch on the conformant planning example**
> > >
> > > This is an example influence diagram (https://drive.google.com/file/d/1SJfFES1jpU4vWgous2qkjmuUOfy15lcC/view?usp=sharing)
> > > of BTUC problem only showing a graphical model part of the problem specification (skipping the description of all the functions, etc). Depending on the way one defines random variables (propositions), other graphical models can also represent the same problem, perhaps better than the one in the link.
> > >
> > > It is also possible to represent the same problem by a grounded RDDL.
> > > @article{sanner2010relational,
> > >   title={Relational dynamic influence diagram language (rddl): Language description},
> > >   author={Sanner, Scott},
> > >   journal={Unpublished ms. Australian National University},
> > >   volume={32},
> > >   year={2010}
> > > }

---

> > > > ### Comment · AnonReviewer2 · 2019-04-11
> > > > **The example clears up things but not completely**
> > > >
> > > > Thank you for adding an example. I strongly recommend to add such an example to the paper as well (as early as possible and maybe with added observations). I still don't follow completely but that could probably be cleared up by a paragraph of text in the paper explaining the example.
> > > >
> > > > While the example helps me understand the setting of the paper, I still cannot follow the main message of the paper. Making this paper accessible to the HSDIP audience would probably require major changes.

---

### Official Review · AnonReviewer1 · 2019-04-05
**Interesting work on reasoning over Graphical Models which may or not be relevant**

**Rating:** 4
**Confidence:** 4

**Review:**

The authors of this work - which generalises the minibuckets inference heuristic by Dechter to influence diagrams, something which I find quite interesting - needs to do a bit more of effort to engage with the literature in planning and its benchmarks.

In the abstract we are told that influence diagrams (ID) can be used as heuristic to guide policy search/policy iteration algorithms for MDPs and POMDPs, and that a comparison with state-of-the-art methods is forthcoming. The former is only true for MDPs which have a finite and known number of stages, referred to as well as finite-horizon MDPs. Otherwise, since the technique relies on unrolling the ID corresponding to each stage of the MDP, it is easy to see that the graph can be infinite unless some notion of a fixed point is given. That is a major technical detail that the authors seem to consider unimportant and they should acknowledge it.

It is also a bit surprising that the definition of an MDP is given in Figure 1. I encourage the authors to review existing formulations and use them to tie clearly influence diagrams and the trajectories induced by policies of an MDP. If I may suggest one such framework, I recommend consulting Dmitri Bertsekas' "Dynamic Programming and Optimal Control", volume 1.

The second part of the promise made by the authors, if abstracts can be considered to be so, does not eventuate either.

The state-of-the-art in factored MDP solving is described in this paper

@inproceedings{trevizan2016heuristic,
  title={Heuristic search in dual space for constrained stochastic shortest path problems},
  author={Felipe Trevizan and Sylvie Thi{\'e}baux and Pedro Santana and Brian Williams},
  booktitle={International Conference on Automated Planning and Scheduling},
  year={2016}
}

and

@inproceedings{trevizan2017efficient,
  title={Efficient solutions for Stochastic Shortest Path Problems with Dead Ends.},
  author={Felipe W. Trevizan and Florent Teichteil-Knigsbuch and Sylvie Thi{\'e}baux},
  booktitle={UAI},
  year={2017}
}

which I don't see being used. Neither less recent, but still interesting heuristic search algorithms like Bonet and Geffner's Anytime-AO* (AAAI 2012) or instances of Monte Carlo tree search like UCT.

The benchmarks themselves include randomly generated problems - which is not necessarily an issue - and domains which have been borrowed from other work. I invite the authors to consider the domains discussed in the paper above.

---

> ### Author Response · Authors · 2019-04-09
> **Thanks for your review.**
>
> Clarification about influence diagram:
> An influence diagram is a graphical representation of a sequence of decision problems combining the following components; Bayesian network capturing the uncertainty of random variables (factored state or chance variables and decision variables), utility functions defining the additive utilities, and the partial ordering constraints (also called information constraints) on the decision (action) variables and chance (state) variables that dictate the precedence of the decisions and the information (history of observed states and decisions) available to the decision maker at each stage.
>
> There are many variants of influence diagram, for a multi-agent game or single agent with limited memory, etc.
> The influence diagram introduced in the paper is the original and the basic one for the single agent sequential decision making with perfect recall (no forgetting).
>
> Finite-horizon MDPs and POMDPs can be represented by an influence diagram and Figure 1 is an example of such a case in a factored MDP. The main idea of using a graphical model (influence diagram) inference is to capture the decomposable structures of a problem (non-serial dynamic programming) and the approximation schemes such as mini-bucket inference heuristic allow additional decomposition by introducing relaxation to the graphical model.
>
> The submitted paper didn't compare against algorithms popular in the planning community, but it compared, favorably,  with other graphical model inference algorithm for solving influence diagrams. Indeed, this is work in progress.  Since Influence diagrams algorithms has a potential for planning we believe we will be able to show their impact. Notice that some earlier works applying methodologies in graphical model inference to conformant planning was shown to have potentials in the literature.
> @inproceedings{lee2016applying,
>   title={Applying Search Based Probabilistic Inference Algorithms to Probabilistic Conformant Planning: Preliminary Results.},
>   author={Lee, Junkyu and Marinescu, Radu and Dechter, Rina},
>   booktitle={ISAIM},
>   year={2016}
> }

---

> > ### Comment · AnonReviewer1 · 2019-04-09
> > **Thanks and some further comments**
> >
> > Thanks for your response, it is much appreciated.
> >
> > It was not my intention to give the impression that I am not expecting an answer. On the contrary, OpenReview offers the opportunity of that and much more. For instance, you should note that you can upload a revision of your paper at your leisure, responding to our comments and critiques in a very compelling form. OpenReview is designed - in my view - as a journal "lite" environment, where reviews aren't just the bombs coming out of the blue that they often are in IJCAI or UAI. This review needs to be read as a "Major Revision" request, rather than an abrupt "Reject".
> >
> > Your reply pretty much outlines already what could be a possible, stronger and much clearer paper that ties in together planning over MDPs where you have models of actions and probability distributions, and optimisation and inference over graphical models.
> >
> > This connection has been pursued several times in the past, mostly led by researchers from the planning community like Joerg Hoffman, Carmel Domshlak, Ronen Brafman, Guy Shani, Blai Bonet and Hector Geffner, just to list the ones that readily come to my mind. Here's a list of references which you may want to study to identify benchmarks and position your work better. Below some bibtex entries from Scholar for the most salient references that I think will help you most
> >
> > @article{domshlak2007probabilistic,
> >   title={Probabilistic planning via heuristic forward search and weighted model counting},
> >   author={Domshlak, Carmel and Hoffmann, J{\"o}rg},
> >   journal={Journal of Artificial Intelligence Research},
> >   volume={30},
> >   pages={565--620},
> >   year={2007}
> > }
> >
> > @inproceedings{somani2013despot,
> >   title={DESPOT: Online POMDP planning with regularization},
> >   author={Somani, Adhiraj and Ye, Nan and Hsu, David and Lee, Wee Sun},
> >   booktitle={Advances in neural information processing systems},
> >   pages={1772--1780},
> >   year={2013}
> > }
> >
> > @article{bonet2014belief,
> >   title={Belief tracking for planning with sensing: Width, complexity and approximations},
> >   author={Bonet, Blai and Geffner, Hector},
> >   journal={Journal of Artificial Intelligence Research},
> >   volume={50},
> >   pages={923--970},
> >   year={2014}
> > }
> >
> > @inproceedings{brafman2014properties,
> >   title={On The Properties of Belief Tracking for Online Contingent Planning using Regression.},
> >   author={Brafman, Ronen I and Shani, Guy},
> >   booktitle={ECAI},
> >   pages={147--152},
> >   year={2014}
> > }
> >
> > @inproceedings{bonet2016factored,
> >   title={Factored Probabilistic Belief Tracking.},
> >   author={Bonet, Blai and Geffner, Hector},
> >   booktitle={IJCAI},
> >   pages={3045--3052},
> >   year={2016}
> > }
> >
> > As you will see from the citation counts and other statistics, these works have not really caught on with the planning community, which in my opinion has concentrated its efforts on figuring out planning over MDPs (partially or fully observable) from a logical perspective. That is, belief tracking, which is the task that inference over an influence diagram I think serves best, rather than producing posterior probability distributions on the values of state variables given a history of observations, produce rather proofs of applicability of actions. They are two different inference tasks, but closely related.
> >
> > I hope you find the above helpful.

---

> > > ### Author Response · Authors · 2019-04-10
> > > **Thanks for the comments**
> > >
> > > We are aware of the works in the provided list.  On the perspective of graphical models, and based on our experience, there are many challenges with the representation of problems; typical planning languages involve many challenging aspects in inference such as first order or relational logic, global constraints, functions defined over other functions (for example, probability value depending on the previous rewards). Each of the components requires special attention by lifted inference, high-order inference, and other possible approaches but all such interesting and challenging things are mixed and come at once in many planning problems that mostly be avoided by sampling or simulation-based approaches. So, it is non-trivial to compare algorithms rooted from different methodologies because algorithms are somewhat tied to representation already.
> > >
> > > we appreciate knowing the relevant works and state-of-the-art methods for solving probabilistic planning problems.

---

> > > > ### Comment · AnonReviewer1 · 2019-04-11
> > > > **Let me unpick some of those arguments**
> > > >
> > > > I am not sure I am getting the intended meaning of your answer. If you were aware of those works, then it is totally on you not having a related work section that puts your potential contribution in context. It also sounds vaguely dismissive, but I don't want to read too much more into that.
> > > >
> > > > You note a few challenges regarding the application of tools from inference in graphical models into planning. Other than pointing to the standing examples of such applications, and in some cases, quite directly present in the list of publications above, I am happy to oblige and go over each of them:
> > > >
> > > > 1/ First order relational logic
> > > >
> > > > While it is true that benchmarks in planning use such representations, in practice, no true first-order representation is used by planners, as there is invariably always a pre-processing step that grounds the first order representation. Also, planning domains, especially for non-classical planning, rarely if ever, use quantifiers, or formulas more complicated that conjunctions of positive and negative literals of atoms.
> > > >
> > > > Alternative representations of probabilistic and partially observable exist, which are actually inspired by work on inference over dynamic Bayesian networks. The one used in the latest probabilistic planning competitions, due to Scott Sanner
> > > >
> > > > http://users.cecs.anu.edu.au/~ssanner/IPPC_2011/RDDL.pdf
> > > >
> > > > for which many tools exist that faciliate enormously parsing
> > > >
> > > > https://github.com/thiagopbueno/pyrddl
> > > >
> > > > and simulating the resulting (PO)MDPs:
> > > >
> > > > https://github.com/thiagopbueno/tf-rddlsim
> > > >
> > > > 2/  global constraints
> > > >
> > > > Global constraints, beyond the obvious one of plan validity (belief tracking in the partially observable setting), are an active area of research in planning. Actually, I personally would expect reasoning over graphical models to offer a comprehensive formal and computational framework to tackle them.
> > > >
> > > > On the other hand, I would also find personally very surprising that work a compelling integration of cutting edge optimisation algorithms for graphical models to be rejected outright because it is "not handling global constraints".
> > > >
> > > > 3/ functions defined over other functions (for example, probability value depending on the previous rewards)
> > > >
> > > > I am not entirely sure what this meant. My best guess is that you're observing that the treewidth of a "direct" unrolling of a DBN-like structure is not bounded, or very high. That's exactly one of the motivators behind the work by Bonet and Geffner on tractable inference.
> > > >
> > > > >  it is non-trivial to compare algorithms rooted from different methodologies because algorithms are somewhat tied to representation already
> > > >
> > > > It is not trivial, if it was probably the question would have little or no scientific interest. I would advise that you look closely at RDDL and realise that we're not talking so much about fundamental issues in representation, as RDDL is pretty much a "Graphical Models" representation.
> > > >
> > > > But actually, and rather than fundamental differences, what we're really talking about is whether or not algorithms require certain constraints to be represented explicitly, e.g.  "frame axioms" that propagate the value of state variables which are "inertial", that is not affected by any action. Other algorithms, typically based in some form of DP, can handle those constraints without representing them explicitly. The later do not suffer any blow up in the size of the computational problem to solve, but there are certain things, like handling well stage and global constraints, that they cannot do well.

---

> > > > > ### Author Response · Authors · 2019-04-11
> > > > > **clarification about  submission.**
> > > > >
> > > > > This submission assumes a problem is given in an influence diagram in the UAI format. The goal of the current work is to come up with an approximate inference that can be used as a static heuristics for search.
> > > > > It is a work in progress that the submission didn't integrate with search and other possible things;
> > > > > this submission only addresses a narrow and specific topic about using graphical model inference to solve probabilistic planning.
> > > > > obviously, there are so many similar yet different formulations are available for the probabilistic planning and we picked up one that is
> > > > > natural in the graphical model framework.
> > > > > It will be interesting future work to do some comparative study with selected planners (mostly search based) on some planning benchmarks.
> > > > > Obviously, this workshop submission does not cover such topics at this point.
> > > > >
> > > > > We believe that we are well aware of the issues of different formats across planners and languages, etc. Thanks for pointing out some resources that we are also aware of. they are not directly compatible with UAI format, so some additional works independent to the generation of heuristic is also going on.
> > > > > this submission does not address such an issue;  it must be a good topic for another workshop in ICAPS about planning competitions.
> > > > >
> > > > >
> > > > > Followings are answers for a few specifics that you point out:
> > > > >
> > > > > 1. On the other hand, I would also find personally very surprised that work a compelling integration of cutting edge optimisation algorithms for graphical models to be rejected outright because it is "not handling global constraints".
> > > > >
> > > > > We believe that integration needs additional efforts on top of the generation of heuristic by the method described in the submission.
> > > > > Btw, if you are aware of some concrete projects that integrating cutting edge optimization algorithms, graphical models, and CP like system handling global constraints I will be happy to learn it.
> > > > > The ones that closest to that came into my mind are a handful of mathematical programming solvers and probabilistic programming languages;
> > > > > still, each solver has its strength and some missing components.
> > > > > Unfortunately, it is beyond scope for us to implement such a system.
> > > > >
> > > > >
> > > > > 2. It is not trivial, if it was probably the question would have little or no scientific interest. I would advise that you look closely at RDDL and realise that we're not talking so much about fundamental issues in representation, as RDDL is pretty much a "Graphical Models" representation.
> > > > >
> > > > > The fact that an RDDL model instance is pretty much like a "Graphical Models" representation does not necessarily guarantee that all graphical model inference and learning algorithms process such an input format. As far as I know, there's no such project available.
> > > > > Of course, it is valid to make a statement that one can write a probabilistic program reflecting the semantics of each RDDL problem domain or even translating the grammar between different languages. The validity of a statement is vague when it comes to an empirical study that requires all the actual implementations.
> > > > >
> > > > > 3. The last paragraph also highlights some interesting aspects and insights of algorithm design over different methodologies. However, the scope is beyond the topic presented in the current submission.
> > > > >
> > > > > I tried to make things brief because this time you said that "I don' want to read too much into that."  Thanks for some inputs that might worth considering.

---

> > > > > > ### Comment · AnonReviewer1 · 2019-04-11
> > > > > > **Thanks for the follow up, time to wrap up the discussion from my side**
> > > > > >
> > > > > > I don't want to read too much into "being aware of X" and then going onto "we are not using X because of reasons" when it is not clear what "aware" actually means. I appreciate the effort of trying to cross over the "lines" separating adjacent fields of research, like planning, optimization and reasoning over graphical models are. But clearly, in this case, I think important facts are being "lost in translation", and I am making an effort to communicate.
> > > > > >
> > > > > > We should generally try to avoid blowing out of proportion software incompatibilities due to "different formats" or confuse them with a problem of incommensurability, like the gap existing between biology and astronomy. What I understand you are pointing at sounds to me like an engineering problem.  In the particular case we are discussing, it is an engineering problem which has been solved by tools you were "aware" of. Those packages have been built using Python, with the express intent of being highly reusable and easy to interface with pretty much everything. As far as I know, it is rare the programming language that cannot interface natively with Python, avoiding the overhead of writing an intermediate file to disk (which is always possible). These are not "vague" remarks, I am sharing a personal experience.
> > > > > >
> > > > > > Of course, if the compilation is blowing up, because say, A uses an implicit/functional representation of conditional probabilities, and B needs explicitly a CPT, then we're looking at something interesting. That is a fundamental limitation of B, whose inference mechanisms need an explicit representation of the relations/functions that make up the graphical model (i.e. graphical model being understood as a graph representation of a CSP, a COP or an unconstrained optimization problem).
> > > > > >
> > > > > > > Btw, if you are aware of some concrete projects that integrating cutting edge optimization algorithms, graphical models, and CP like system handling global constraints I will be happy to learn it.
> > > > > >
> > > > > > Go no further than recent work by the author of the two software packages you were already aware of
> > > > > >
> > > > > > https://github.com/thiagopbueno/tf-mdp
> > > > > >
> > > > > > I think that gradient-based optimization over stochastic computation graphs is quite cutting edge, and involves graphical models. Indeed, it is weaker when it comes to handling state/stage and trajectory constraints (see discussion above regarding explicit vs. implicit models). Other work exists that integrates SMT solvers and convex programming to obtain stunning results that make, in my opinion, very popular sampling-based approaches to planning all but obsolete. But I don't want to bore you reciting work you may be "aware of".
> > > > > >
> > > > > > Just to wrap up the discussion and to make sure I am conveying the right message:
> > > > > >
> > > > > >  - The work presented is interesting (I was, generally speaking, "aware of" it and actually enjoyed reading it) and I can see the relevance it can have to advance the state of the art in basic problems interesting to the planning community (belief tracking, heuristics for SSPs, etc.)
> > > > > >  - It is my opinion that the authors (not the readers, or the reviewers, or the persons attending the workshop) need to do more of an effort to present and articulate their work in a way that their actual and potential contributions can inform, inspire or seem relevant to researchers attending the workshop and the wider ICAPS conference.

---

> > > > > > > ### Author Response · Authors · 2019-04-12
> > > > > > > **Thanks again for all the comments.**
> > > > > > >
> > > > > > > Thanks again for all the comments. We really appreciate all the detailed comments, references and discussions.

---

### Meta-Review · Program_Chairs · 2019-04-23

**Recommendation:** Reject
**Confidence:** 5

**Metareview:**

Thank you for submitting your paper to the HSDIP workshop.

Unfortunately, we cannot accept the paper in its current form.
As noted in one of the reviews, the paper is very challenging for
readers without the necessary background on influence diagrams
and graphical models, which we can expect will be the majority of
the workshop's attendees; to build a bridge between communities,
the paper needs to have more consideration for the understanding
and terminology of the target audience. This year's workshop
program is also very time-constrained, with presentations of 15
minutes or less. (If we had had a more relaxed schedule, a longer,
tutorial-style, presentation on IDs could have been suitable.)

There is certainly a relationship between IDs and (finite-horizon)
stochastic or POMDP planning, and it would be worthwhile exploring
the relation between solution methods from the two areas. The
reviewers have given a number of suggestions for improving the
cross-community readability of the paper, such as including
examples of planning problems formulated as IDs, relating the
technical content to state-of-the-art algorithms for factored
(PO)MDP solving - whether that is by empirical comparison, by
drawing out the similarities and differences in underlying
representations or methods or in some other way - and demonstrating
how the derived bounds can be used as heuristics in search-based
(PO)MDP planner, as mentioned in the paper's abstract.

The ICAPS conference offers a number of ways to introduce the
planning community to work on solving IDs, besides regular paper
submissions. For example, ICAPS typically admits short (1/4-day,
or 90 minutes) tutorials, which can introduce the audience to a
related area. The Journal Presentation Track (cf.
https://icaps19.icaps-conference.org/calls) is another window
through which work on related topics can be introduced to the
planning community.